# Building Floorplan Reconstruction Based on Integer Linear Programming

**Qiting Wang [1,2], Zunjie Zhu [2,3,\*], Ruolin Chen [1,2]** , **Wei Xia [2,4] and Chenggang Yan [1]**

[1] School of Automation, Hangzhou Dianzi University, Hangzhou 310018, China
[2] Lishui Institude of Hangzhou Dianzi University, Lishui 323000, China
[3] School of Communication Engineering, Hangzhou Dianzi University, Hangzhou 310018, China
[4] School of Information and Communication Engineering, Hainan University, Haikou 570228, China
**\*** Correspondence: zunjiezhu@hdu.edu.cn

**Abstract:** The reconstruction of the floorplan for a building requires the creation of a two-dimensional floorplan from a 3D model. This task is widely employed in interior design and decoration. In reality, the structures of indoor environments are complex with much clutter and occlusions, making it difficult to reconstruct a complete and accurate floorplan. It is well known that a suitable dataset is a key point to drive an effective algorithm, while existing datasets of floorplan reconstruction are synthetic and small. Without reliable accumulations of real datasets, the robustness of methods to real scene reconstruction is weakened. In this paper, we first annotate a large-scale realistic benchmark, which contains RGBD image sequences and 3D models of 80 indoor scenes with more than 10,000 square meters. We also introduce a framework for the floorplan reconstruction with mesh-based point cloud normalization. The loose-Manhattan constraint is performed in our optimization process, and the optimal floorplan is reconstructed via constraint integer programming. The experimental results on public and our own datasets demonstrate that the proposed method outperforms FloorNet and Floor-SP.

**Keywords:** point cloud; normalization; planes; Manhattan; floorplan reconstruction

## 1. Introduction

Nowadays, automatic modeling technology is increasingly utilized in diverse tasks such as object reconstruction [1], human pose estimation [2], hand tracking [3,4], and scene reconstruction [5,6]. For the evaluation and analysis of house layout and indoor furnishings, the accurate reconstruction of the overall architectural floorplan of the house is essential. The traditional methods of reconstructing the indoor 2D floorplan of houses apply manual measurement to obtain the dimensional information of each part of the house and then draw the 2D floorplan of the house with computer software. Thus, these methods are time consuming and require high labor costs in practical application. To improve the efficiency of floorplan reconstruction, automated architectural floorplan reconstruction has received primary support in recent years.

One main branch of automated architectural floorplan reconstruction is based on indoor structuring elements. In particular, the existing methods [7] of this branch first segment and classify the main structures of the interior scene, such as ceilings, floors, walls, and some other small architectural elements, from the input mesh data. Then, the segmented indoor structuring elements are employed to reconstruct the architectural floorplan. In the segmentation phase, Random Sample Consistency (RANSAC) plays an important role in extracting the main structures of the house from the point cloud data. Due to the missing parts of the point cloud and the occlusion caused by indoor objects, employing the extracted structural elements is difficult to fully reconstruct the complete indoor layout. To address information deficiency of floorplan reconstruction, a general strategy which exploits the wall structure as the main estimation object and the

contours of the ceiling and floor as prior information is proposed to complete and floorplan reconstruction. Finally, the reconstructed structural elements are assembled into a complete room layout.

As stated in the above analysis, automated indoor layout estimation [8–15] utilizes the detection results of RANSAC to reconstruct the fitted objects and combine them. There are two problems with the approach. On the one hand, automatic indoor layout estimation relies on the detection results and the detection effect of Random Sample Consistency requires high-quality point cloud data. When the point cloud is incomplete, the inaccurate plane will directly lead to the failure of floorplan estimation. On the other hand, the RANSAC-based methods [8,16–21] are less robust to different building scenes. These methods have problems such as difficulty in detecting planes stably, and the inability to adaptively adjust parameters according to the size of the scene. These problems lead to incomplete reconstruction results of the floorplan and the existence of noisy line segments in the floorplan. Therefore, we designed step of plane merging to avoid the plane being divided into multiple zones, and designed integer linear programming to optimize the missing and duplicate problems in RANSAC detection.

In this paper, we first propose a preprocessing method for the input data. The dense point cloud is resampled based on the mesh slices, and each triangular patch is sampled with a uniform probability to generate continuous dense point cloud data. After preprocessing, the point cloud distribution is more uniform, making the plane detection result more accurate. Then, we utilize RANSAC to obtain 3D principal plane analysis results. The proposed method fits the point cloud of the main plane in the 3D scene to obtain the corresponding 2D projected line segment. After obtaining all the candidate line segments, the proposed method constructs a loose-Manhattan global energy optimization method, and filters out the optimal line segment solution set using the energy function to reconstruct the two-dimensional floorplan. In addition, we present a large dataset, called GibLayout, that contains multiple types of interior building models. GibLayout is characterized by several aspects. Firstly, the large-scale dataset contains 80 models with room numbers from 3 to 11. Each model contains RGBD images, point cloud models, and a groundtruth floorplan. The architectural scenes in the dataset are different, which allows the method to be sufficiently tested and further improves the robustness of the method. Secondly, we manually annotated the location information of walls and doors in the model of the dataset based on point clouds. It lays a solid foundation for the wide application of the dataset in the field of indoor layout estimation. More detailed descriptions are provided in Section 4.

In summary, our contributions are:

1. We propose an indoor floorplan reconstruction framework, which normalizes the inputs via mesh-based resampling, and extracts walls from candidates by using the proposed energy function on 2D planes.
2. We propose a near-Manhattan global energy optimization method, which tightly combines the geometric information of the point cloud and uses the energy function to solve the optimal solution set.
3. We release a large floorplan benchmark named GibLayout, which contains 3D models of 80 indoor scenes with more than 10,000 square meters.

This paper is structured as follows. The related works are introduced in Section 2. Section 3 explains the framework of our approach to estimate interior layouts. Experiments and the discussion are presented in Section 4. Finally, conclusions are drawn in Section 5.

## 2. Related Works

In recent years, research on the reconstruction method of the floorplan has attracted extensive attention and still is a current topic of ongoing work. Below, we summarize a number of methods used to deal specifically with indoor building reconstruction.

Some methods are based on structuring element extraction to achieve the reconstruction of 2D floor plans. Sanchez et al. [22] proposed a method for the 3D modeling of buildings from point clouds to planes. The method divides indoor point clouds into ceil-

ings, floors, walls, and other small building structures through a Principal Component Analysis (PCA) algorithm, and then employs Random Sample Consistency to fit the detected planar primitives. Okorn et al. [23] project the wall plane structure obtained by plane detection onto the horizontal plane and perform wall segment detection based on Hough transformation to model a two-dimensional floor plane. Shi et al. [24] propose an approach using semantic geometric modeling to reconstruct 3D architectural models using semantic information from unstructured 3D point clouds. This method improves the accuracy of detecting wall surfaces. Budroni et al. [25] propose techniques for the fully automated 3D modeling of indoor environments from point clouds acquired through multiple scans, which are then processed to segment planar structures with obvious architectural significance such as floors, ceilings, and walls. Adan et al. [26] present a method for using laser scanner data to predominantly model planar surfaces which could identify candidate surfaces for modeling, label occluded surface regions, and detect openings in each surface using supervised learning and reconstruct the surface in the occluded regions. Xiong et al. [27] present a method to automatically convert the raw 3D point data from a laser scanner positioned at multiple locations throughout a facility into a compact, semantically rich information model with their method focused on recovering detailed surface labellings.

Some approaches achieve reconstruction by spatially segmenting the entire building. Room segmentation can pre-filter a large number of outliers, improve performance and accuracy, and the properties of rooms can also be fully demonstrated. The segmentation of the room is performed to exploit the physical and geometric characteristics of the data to merge the similar parts observed at the same location and detect the room based on visibility. Mura et al. [10] proposed a room detection link in the reconstruction process. A room is defined as a collection of polygonal areas on a plane. Using distance calculation, and iterative clustering, extracting a new room as a cluster of polygonal regions in each iteration can be achieved. Each scan location is located in a room. Each room is scanned from at least one location within its boundaries, and the clustering terminates when each scan location is assigned to a room cluster. Ochmann et al. [20] proposed a method for segmentation based on detecting 2D regions of building planes. The method assumes one-to-one mapping between input models and rooms and assigns a room label to each 2D region.

MURA et al. [28] use Markov random clustering to obtain the correct number of rooms before room reconstruction, which can be used to extract rooms using label optimization. Clustering the overlapping positions during scanning can effectively avoid merging these positions later. Mura and Pajarola [29] proposed placing a set of larger view centers in the model, taking them as scan points, building centers of the leaf cells of an adaptive octree, and then using visibility-based clustering to extract room methods. OCHMANN et al. [17] proposed a method to segment the point cloud into small patches and using the center of the patch as the view detection point. By computing synthetic viewpoints, avoiding the requirement for input scan positions, viewpoints provide label information for optimization of the generated room model.

In recent years, deep learning has made remarkable progress in the task-image domain [30–32] . As an advanced method based on deep learning, FloorNet [33] was proposed as a deep neural architecture. Using a single image as input, three network branches predict pixels' geometric and semantic information, namely (1) a single-layer PointNet [34] to obtain 3D information, (2) a top-down view of 2D point density images to enhance local spatial inference, (3) a full image information input to a convolutional neural network (ConvolutionNeuralNetwork, CNN) to generate deep image features. By exchanging intermediate features across branches of FloorNet, RGB-D video with a camera bit pose is converted into pixel planar geometric and semantic information, and a planar map satisfying the constraints is obtained by integer programming. Additionally, at the same time, it can recover the semantic labeling information of the house in the 2D floor plan, such

as a part of the floor area belonging to the bedroom, kitchen, hallway, etc. However, the method is only applicable to the scenario under the Manhattan assumption.

In 2019 cheng et al. proposed another Floor-SP [9] method based on FloorNet, which optimizes the floor plan structure by solving the shortest path problem in descending order of room coordinates. The objective function contains a data term, which is guided by a deep neural network, a continuous term to express the consistency term of adjacent rooms sharing corners and walls, and a model complexity term. Unlike most other methods, this method does not require corner point or edge extraction and does not require Manhattan constraints. However, the MaskRcnn [35] network used in this method for room clustering segmentation masks a single room with a large BoundingBox, resulting in a larger detected room area than the original room, thereby making the reconstructed floor plan house edges and corner point information incompatible with the true value. At the same time, this method requires very strict data pre-processing, and the model is orthogonally aligned with the coordinate system before it can accurately and quickly segment and cluster the houses.

Phalak et al. proposed the method Scan2Plan [36] in 2020, which clusters rooms and walls individually by PointNet and then calculates the perimeter of a room wall to complete the reconstruction of the floor plan. However, this method is only applicable to simple small scene reconstruction and cannot recover the floor plan structure of small rooms inside the scene; rather, only the outer contour of the scene can be reconstructed.

Furthermore, research on collecting building information with the help of remote sensing platforms to extract building outlines has also continued [37–40]. The study provided in [37] investigates the characteristics of small UAS and their impact in the context of remote sensing models, identifying new remote sensing capabilities and the challenges posed by such platforms. It is instructive that the affordability and potential of the ubiquitous operation of small UASs have led to an increase in the type and quality of building information that can be collected so that such remote sensing systems can be applied to solve information collection problems [37].

Existing mathematical methods have also been widely applied in the field of 2D floor plan reconstruction in recent years, and have shown outstanding results.The proposed vec-20 [7] reconstructs floor plans by using energy function. The method takes the building 3D point cloud data as the original input, detects the wall data with the RANSAC [19] method and projects them into 2D line segments, and selects the optimal set of segments to compose the floor plan by optimally solving the energy function using the energy function. However, the current RANSAC-based method is less robust to different building scenes, and it has problems such as difficulty in detecting small planes stably and the inability to adjust the parameters adaptively according to the scene size, which lead to the problems of incomplete floor plan reconstruction results and the existence of noisy line segments. Yaxin Li et al. [41] proposed an automatic indoor AB BIMs generation framework by using low-cost RGB-D sensors in 2020. Fan Yang et al. [42] proposed a straight and curved line tracking method followed by a straight line test. Robust parameters were used, and a novel straight line regularization method was achieved using constrained least-squares in 2019. Maarten Bassier et al. [43] propose a connection evaluation framework and creates a logical BIM model in an unsupervised manner. However, since the method has not been compared and the method cannot be compared with open source, this paper only compares the Floor-SP and FloorNet methods.

## 3. Proposed Method

In this section, we introduce the framework of our automatic indoor floorplan reconstruction system, for which the framework can be viewed in Figure 1. Firstly, we perform 3D model normalization using mesh-based re-sampling (Section 3.1), which normalize the 3D models from different datasets or sensors to ensure the robustness of our method. The output of this step is a point cloud of the indoor scene. Then, we utilize normal vector information to preserve vertical walls. In addition, we project the point cloud data of the vertical wall into a 2D plane to form a set of line segments. Finally, the closed-loop walls

are optimized using the energy function with geometry and Manhattan constraints, and the floorplan is acquired by the constraint integer programming solver (Section 3.3).

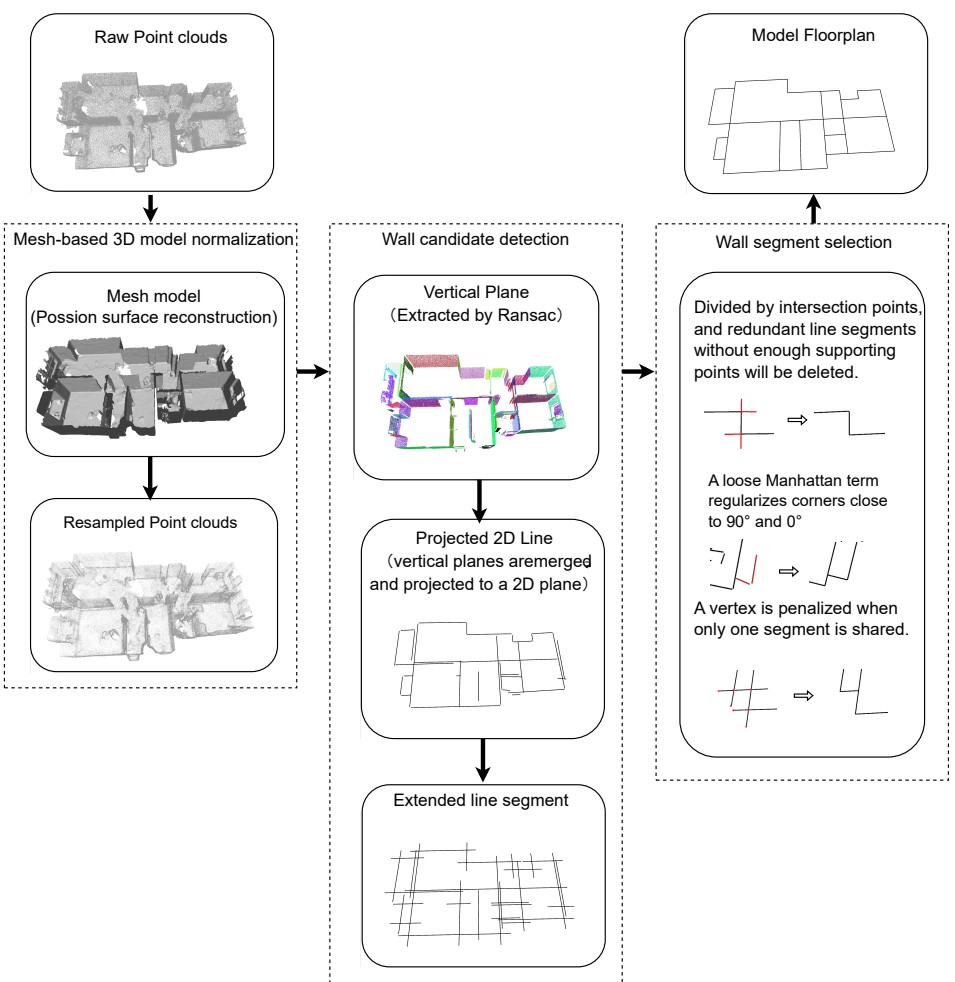

**Figure 1.** Overview of the proposed method. The input is a mesh model and the output is a floorplan.

### 3.1. Mesh-Based 3D Model Normalization

The non-uniform factors of different point clouds, including noises, density, pose and so on, result in the instability of floorplan reconstruction. To ensure the robustness of our method, we transform the point cloud of an indoor scene into mesh representation with the Poisson surface reconstruction method [44]. We normalize the pose of the scene model by paralleling the normal of the scene floor with the z-axis of the world coordinate. As the floor and cell are much larger than other planes in a house, we detect the floor and cell according to the point number of planes.

Then, we generate a point cloud with uniform density by re-sampling the mesh model. As shown in Figure 2, for a triangular patch $M$, a 3D point $\mathbf{p}$ on the path is re-sampled as follows. Firstly, the point $\mathbf{O}$ is randomly selected from the three vertices, and the two adjacent edges $\mathbf{OA}$ and $\mathbf{OB}$ are set as the basis vectors. Then, $r, d$ are generated randomly from 0 to 1 with the constraint of $r + d < 1$. Finally, the 3D point $\mathbf{p}$ is acquired as follows:

$$\mathbf{p} = O + r \cdot \mathbf{OA} + d \cdot \mathbf{OB}. \tag{1}$$

For a triangular patch with area $S_M$, the amount $N_M$ of re-sampling points is derived as follows:

$$N_M = \Theta\left(\frac{S_M}{S^*}\right), \tag{2}$$

where $S^*$ is the hyper-parameter that decides the density of a re-sampled point cloud, and $\Theta$ is the pre-set amount of re-sampled point clouds. A uniformly distributed point cloud is acquired after re-sampling points for all patches.

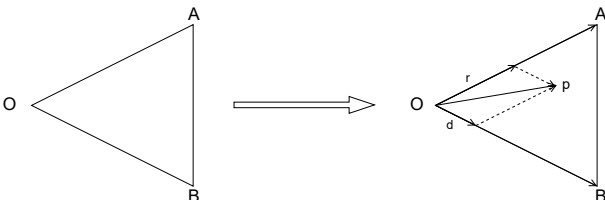

**Figure 2.** Random point re-sampling based on triangular mesh.

### 3.2. Wall Candidate Detection

To reduce the influence of indoor furniture on plane detection, we discard points near the floor by assuming as the interior furniture is always on the floor. Then, we detect planes from the preprocessed point cloud using the Random Sample Consensus (RANSAC), and only vertical planes are selected as candidates of walls. Large planes are separated to multiple small ones in RANSAC. Inspired by Han et al. [7], we propose a method for plane merging. We recombine wall candidates when the following two conditions are reached:

1. The angle between the normals $\mathbf{n}_i$ and $\mathbf{n}_j$ of two planes is less than $\theta$.
2. The distance between the two planes is less than $d$.

The normal vector $\mathbf{n}$ of the merged new plane is calculated as follows:

$$\mathbf{n} = \frac{\mathbf{n}_i + \mathbf{n}_j}{\|\mathbf{n}_i + \mathbf{n}_j\|} \tag{3}$$

Reconstructing the floorplan in the 2D space is more computationally efficient and robust than in the 3D space. To generate 2D walls from the 3D point cloud, we project the points of 3D wall candidates and fit 2D wall segments according to the projected 2D points. Due to the problems of the model itself and the lack of point cloud data in the projection process, some fitted line segments cannot intersect, and corners cannot be closed. Therefore, we extend the segments to intersect the nearby segments according to the distance and intersectability of the endpoints and other segments, and acquired a new segment set $\mathbf{S}_i$. The detailed implementation of the algorithm is presented in Algorithm 1. Figure 1 shows the result of extending 2D segments.

### 3.3. Wall Segment Selection

After obtaining the 2D segment set $\mathbf{S}$ and the vertices of all segments $\mathbf{V}$, each vertice is shared by two or more segments and the connection matrix between vertices and segments is $\mathbf{M}$. We detected wall segments by minimizing the energy function $E$, and three terms were proposed to form the function, including the Manhattan term $M$, the confidence term $C$ and the topology term $L$:

$$E = \lambda_1 \cdot M + \lambda_2 \cdot C + \lambda_3 \cdot L, \tag{4}$$

where $\lambda_1$, $\lambda_2$, $\lambda_3$ are balance parameters.

The Manhattan term in Equation (4) is used to exclude the segments that are not orthometric:

$$M = \frac{1}{N} \cdot \sum_{i=1}^{N} \left( \sum_{\mathbf{S}_j \in N_{\mathbf{S}_i}} |\Psi\{\mathbf{S}_i, \mathbf{S}_j\}| \cdot s_j \right) \cdot s_i, \tag{5}$$

where $N$ is the amount of segments, and $\mathbf{S}_j \in N_{\mathbf{S}_i}$ are segments that intersect with $\mathbf{S}_i$. $\Psi(\mathbf{S}_i, \mathbf{S}_j)$ is the cosine of the vectors of $\mathbf{S}_j$ and $\mathbf{S}_i$. The bool vector $\mathbf{s}$ represent the existence of each segment.

---

**Algorithm 1** Wall Merge and Refining

---

**Input:** $wall_{Ransac}$: Set of walls obtained by Ransac; $\theta$: Angle threshold; $d$: Distance threshold; $MaxD$: MaxExtendThreshold; $Wall_{Dealed}$: Fitted 2D line segment;

**Output:** Merged Wall Collection;

 1: set $x_0 = x_{best}$, compute initial energy function $E(x_0)$;
 2: **for** $i = 1; i < wall_{Ransac}; i{+}{+}$ **do**
 3:    **for** $j = 1; j < wall_{Ransac}; j{+}{+}$ **do**
 4:       **if** i == j **then**
 5:          continue
 6:       **end if**
 7:       Calculate the distance between planes: Distance = Dis($n_i$, $n_j$)
 8:       Calculate the angle between planes: Ang = A($n_i$, $n_j$)
 9:       **if** Ang $< \theta$ and Distance $< d$ **then**
10:          Construct n using Equation (3);
11:       **end if**
12:    **end for**
13: **end for**
14: **for** $i = 1; ; i{+}{+}$ **do**
15:    **if** Ang = A($n_i$, $n_j$) $< 45°$ **then**
16:       $wall_i = wall_{Horizontal}$;
17:    **else**
18:       $wall_i = wall_{Virtical}$;
19:    **end if**
20: **end for**
21: **for** $i = 1; i < wall_{Horizontal}; i{+}{+}$ **do**
22:    **for** $j = 1; j < 2; j{+}{+}$ **do**
23:       **for** $k = 1; k < wall_{Virtical}; k{+}{+}$ **do**
24:          Calculate the distance Point2WallDistance between point and $wall_{Virtical}$[k]:
25:          Point2WallDistance = Dis($wall_{Horizontal}$[i][j], $wall_{Virtical}$[k])
26:          **if** Point2WallDistance $< MaxD$ **then**
27:             **if** $wall_{Virtical}$[k] is SameSide **then**
28:                Extend $wall_{Horizontal}$[i][j]
29:             **end if**
30:          **end if**
31:       **end for**
32:    **end for**
33: **end for**

---

The confidence term is to punish the segments without enough 3D points:

$$C = 1 - \frac{1}{N} \cdot \sum_{i=1}^{N} \left( \min\left\{ 1, \frac{N_{\mathbf{S}_i}}{\delta \cdot L_i} \right\} \cdot \mathbf{s}_i \right), \tag{6}$$

where $L_i$ is the length of the segment $\mathbf{S}_i$, and $\delta$ is a fixed density value that we define per unit length should have with $\delta$ points. $N_{\mathbf{S}_i}$ is the closeness of the projected point to the segment $\mathbf{S}_i$, i.e., the distance between the projected point and the segment is less than the threshold $\sigma$. In our experiment, the $\sigma$ is set as 0.05 m.

The topology term in Equation (4) refines segments by judging the rationality of vertices. $\mathbf{v}$ is an $N_v * 1$ bool vector that denotes the whether a vertice is incorrect, e.g., $\mathbf{v}_i = 1$ when the *i*-th vertice is only shared by one segment. $N_v$ is the amount of all vertices. So,

$$\mathbf{v}_i = \begin{cases} 0, & if \quad \mathbf{M}_i \cdot \mathbf{s} \neq 1 \\ 1, & if \quad \mathbf{M}_i \cdot \mathbf{s} = 1 \end{cases}, \tag{7}$$

where the size of the connection matrix **M** is $N_v * N_s$. $N_s$ is the amount of all candidate segments. $\mathbf{M}_i$ is the $i$-th row of **M**. Therefore, $\mathbf{M}_i * \mathbf{s}$ is the amount of segments that share the $i$-th vertice. Then, the topology term is

$$L = \sum_{i=1}^{N_v} \mathbf{v}_i, \tag{8}$$

Finally, the energy function in Equation (4) is optimized by the Constraint Integer Programming solver [45], and the floorplan is composed of all valid wall segments with $\mathbf{s}_i = 1$.

## 4. Experiments

In this section, we introduce the datasets utilized in our experiments, including the GibLayout Dataset proposed in this work as well as the dataset proposed in FloorNet [33]. We compare our proposed method to the SOTAs qualitatively and quantitatively with the evaluation metrics proposed in the CVPR BIM Challenge. Finally, we perform an ablation study to further evaluate the performance of the proposed Manhattan term of our method.

### 4.1. Datasets and Evaluation Metrics

We introduce two datasets for the evaluations in our experiments, namely our GibLayout Dataset and the FloorNet Dataset. Three evaluation metrics are measured for each of the methods, including Precision, Recall, and the Betti error.

#### 4.1.1. Dataset

We propose a new dataset GibLayout Dataset (our dataset is released in "https://github.com/W-Q-T/Giblayout" (accessed on 1 March 2022)) for the task of floorplan reconstruction, which contains 80 house models with 3 to 10 rooms in each model, and the total area of 80 models is more than 10,000 square meters. In our dataset, the point clouds of house models is selected from the public Gibson Dataset (For more details visit gibsonenv.stanford.edu/database (accessed on 1 November 2021)) [46] collected by NavVis (For more details visit https://www.navvis.com/zh/vlx (accessed on 1 November 2021)) which utilizes two multi-layer LiDAR sensors to collect 3D measurements, combined with industry-leading SLAM algorithm software, to generate mapping-quality point clouds. The obtained point cloud accuracy is 6 mm in a 500 square meter test environment. The laser wavelength of the device is 903 nanometers and the range is 100 m. The number of points collected by the device per second is $2 \times 300,000$. The dataset was collected in 2018 and it covers a diverse set of spaces, e.g., offices, garages, stadiums, grocery stores, gyms, hospitals, and houses. Since we utilize the mesh model from the dataset, we evaluate the model by introducing the specific surface area (SSA), which represents the ratio of the surface area and volume of the inner mesh to the convex hull of the mesh. This is a measure of the level of clutter in the model. The specific surface area of the Gibson dataset is 1.38.

The Gibson Environment database is collected from real indoor spaces and provides each building with a corresponding 3D mesh model, RGB panorama and camera pose information. We layer the house Mesh model provided by the Gibson Dataset, and then resample the single-layer mesh model utilizing the method proposed in Section 3.1. The constant resolution for the resampling is 1000 points per square meter on the surface of the mesh. The groundtruth (GT) floorplan of the dataset is based on these point clouds.

The groundtruth (GT) floorplan of each model is artificially annotated with the professional tool CloudCompare [47]. The GT floorplan contains the labels of walls and doors, where each one consists of two vertices. For example, the two vertices of a wall are labeled by finding the corners in the point cloud. To acquire accurate labels, we project walls point clouds to the floor and then find corners accurately with the guidance of the mask with 0.5 cm grids. The system error of the annotated GT floorplan is smaller than 2 cm, which is adequate for the evaluation of floorplan reconstruction. As shown in Figure 3,

to evaluate the resulting dataset and its accuracy, we project the point clouds to a 2D plane after removing the ceiling and floor and compare it to the GT floor plan overlay. The figure shows that the distance error between the GT floorplan and point clouds is within 2 cm. We separate the Giblayout dataset into training and test sets with 40 models, respectively, and our dataset has been released.

We also utilize the open-source dataset of FloorNet [33] for testing. The 3D models of FloorNet Dataset are collected by the smartphone Google Tango, and the GT floorplans are also generated manually.

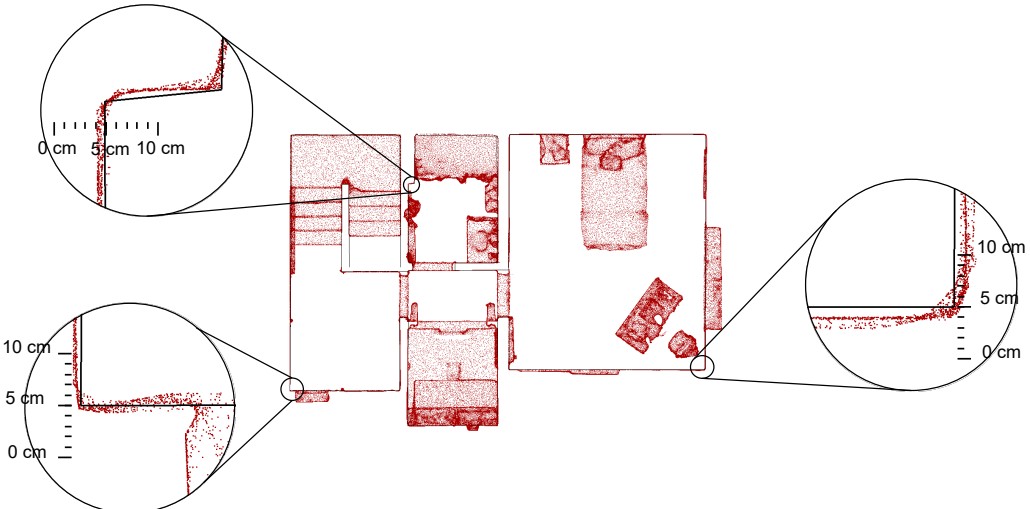

**Figure 3.** Comparison of the annotated GT floorplan and wall point clouds. The red part represents the projection of the point cloud on the 2D plane after removing the ceiling and floor, and the black line represents the annotated GT floorplan.

4.1.2. Evaluation Metrics

We use geometric and topological metrics for the evaluation, including the precision and recall of vertices, and the Betti number error between the GT and the reconstructed floorplan. To measure the accuracy of a reconstructed floorplan, we calculate the precision and recall of reconstructed vertices by performing a comparison with the positions of matched GT vertices. The precision and the recall is formulated as follows:

$$\text{Precision} = \frac{P_{true}}{P_{res}}, \tag{9}$$

$$\text{Recall} = \frac{P_{true}}{P_{gt}}, \tag{10}$$

where $P_{true}$ is the amount of vertice pairs that leads to the distance between the GT and the estimated vertices being smaller than the threshold, $P_{res}$ is the amount of vertices in the reconstructed floorplan. $P_{gt}$ is the amount of vertices in the ground truth floorplan.

Then, we calculate the Betti number error to evaluate the similarity of the room connectivity between the reconstructed floorplan and that of the GT. We compare the sampled patches from each floorplan, and the Betti Error is calculated as follows:

$$\text{Betti number error} = \frac{1}{N_D} \sum_{i=1}^{N_D} |D_{Gt} - D_{result}|, \tag{11}$$

where the Betti number error compares the Betti numbers between the prediction and the ground truth and outputs the absolute value of the difference. $D_{Gt}$ is the Betti number of the floorplan in the GT, $D_{result}$ is the one of the reconstructed floorplans, and $N_D$ is the sampling time. In our experiment, the $N_D$ is 500.

## 4.2. Implementation Details

All experiments were performed on a platform with a Nvidia GTX 1660 GPU and a 12-core Intel-XEON CPU. We generate the normalized point cloud of each model using the random number $r, d$ in Section 3.1. Weights in Section 3.3 strike a balance between the model completeness and complexity. In our experiments, we set $\lambda_1 = 0.5$, $\lambda_2 = 0.2$, $\lambda_3 = 0.3$ for both the FloorNet dataset and ours.

## 4.3. Result and Analyze

In this section, we compare our method with FloorNet [33] and FLoor-SP [9] on the two datasets, and a qualitative and quantitative evaluation are performed.

### 4.3.1. Evaluation on the FloorNet Dataset

FloorNet consists of three networks, the inputs are a Manhattan-corrected point cloud and a 2D room semantic map output from a semantic segmentation network. Floor-SP takes the point density/normal image in the top-down view as the input, and it also utilizes the network to obtain both the room segmentation result and the corner/edge likelihood.

Figure 4 shows the qualitative results of each method. It shows that our method reconstructs in greater detail than Floor-SP and FloorNet.

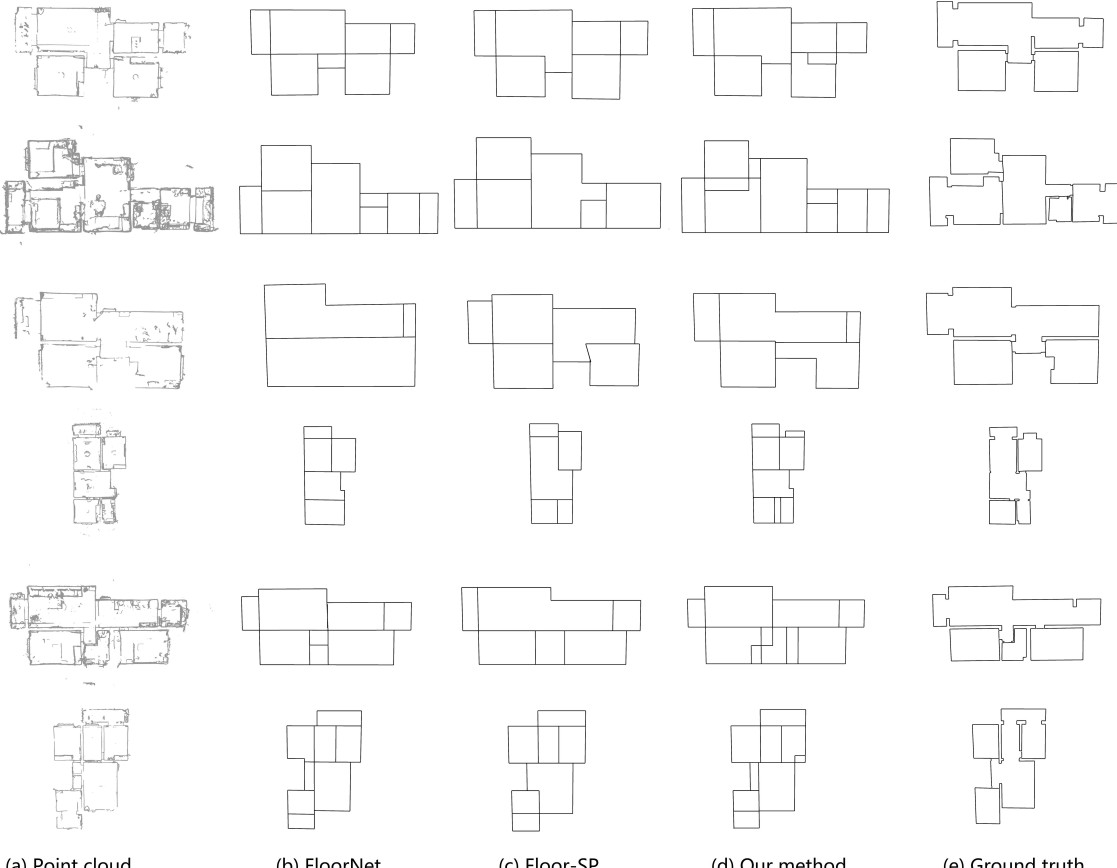

(a) Point cloud     (b) FloorNet     (c) Floor-SP     (d) Our method     (e) Ground truth

**Figure 4.** 2D floorplan reconstruction results of our method and learning-based methods for evaluated datasets. From left to right, we display the density map of the point cloud, the result of FloorNet, Floor-SP, and our method and ground truth, respectively.

FloorNet requires room label information as an additional input for auxiliary detection and reconstruction. Incorrect label coordinates cause the room position to deviate entirely from the correct label coordinates during reconstruction. Floor-SP relies on relatively good semantic segmentation results for the optimization of the room structures. Therefore, the results of the Floor-SP depends on the generalization of the segmentation network.

Furthermore, the intermediate process of Floor-SP and FloorNet converts the floorplan into an image, which reduces the accuracy of the house structure. Both methods utilize small resolution maps (256 × 256), which makes it difficult to capture structural details and reduces the quality of the final result.

Different from these two methods, we utilize the geometric information of the point cloud to transform the 3D model reconstruction problem into an optimization problem in 2D space, which enhances its robustness against noise and missing regions.

As shown in Figures 5–7, the proposed method outperforms the other two methods in terms of recall, precision, and the Betti error. On the one hand, Floor-SP takes the top-down view as input, uses the network to obtain room segmentation results, and iteratively searches for the shortest path for each room. In contrast, planes and line segments fitted from point clouds are more accurate. Floor-SP clusters different rooms by mask, while ignoring the connection between each room, which makes the detail processing of room connections poor. After the clustered data are reorganized, they cannot conform to the topological structure of the original data.

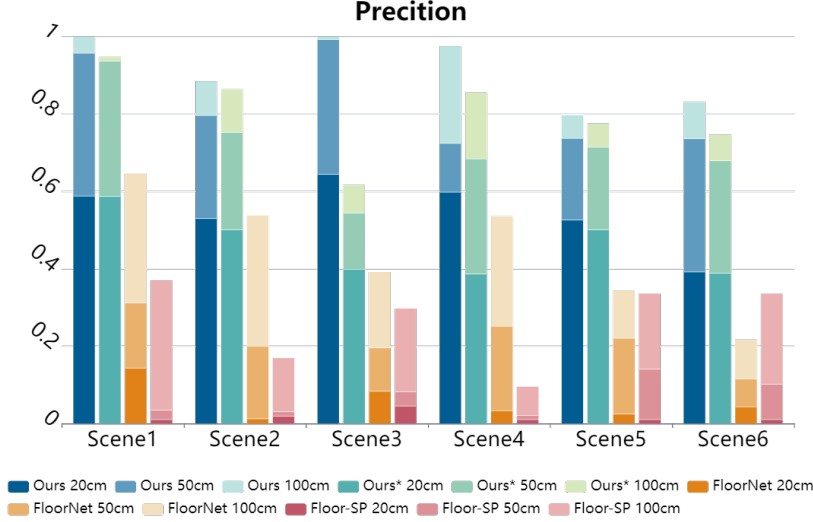

**Figure 5.** Quantitative comparison of our method with FloorNet and Floor-SP in terms of precision. The figures are represented by increasing numbers. Our* is the non-Manhattan result of our method.

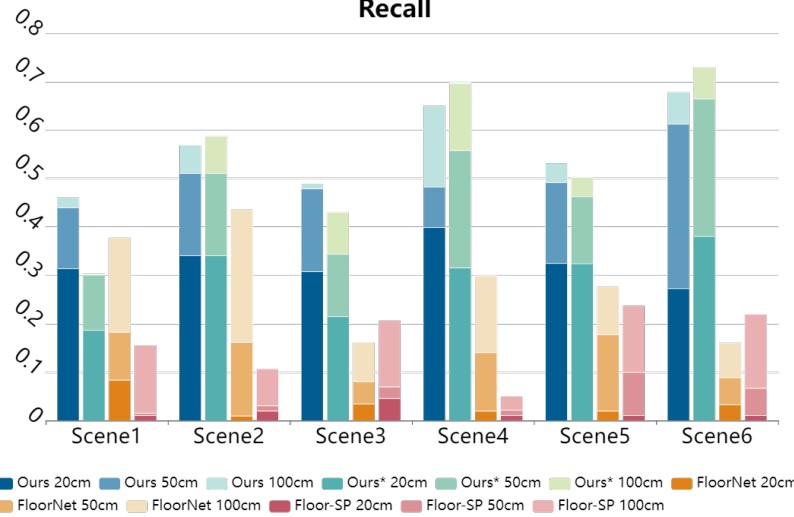

**Figure 6.** Quantitative comparison of our method with FloorNet and Floor-SP in terms of recall. The figures are represented by increasing numbers. The figures are represented by increasing numbers. Our* is the non-Manhattan result of our method.

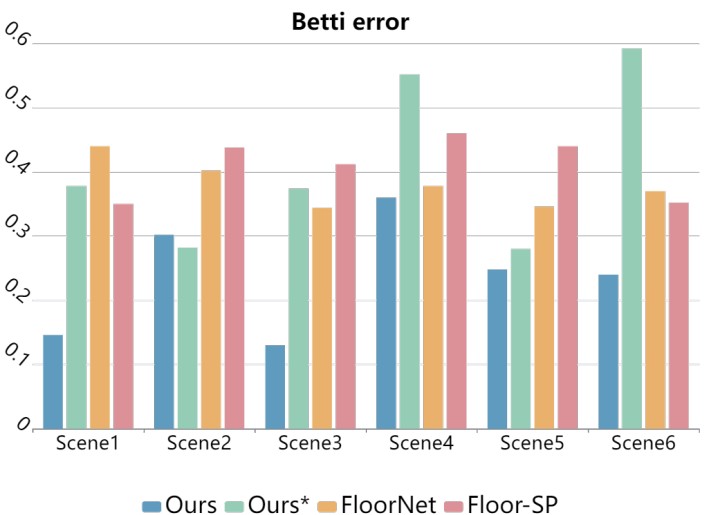

**Figure 7.** Quantitative comparison of our method with FloorNet and Floor-SP in terms of Betti error. The figures are represented by increasing numbers. Our* is the non-Manhattan result of our method.

The energy function optimization method proposed in this paper constrains the topological structure of the house, and there is no twisted structure in the edge profile of the house, as the houses are tightly connected without any gaps.

### 4.3.2. Evaluation on the GibLayout Dataset

In this section, we qualitatively and quantitatively evaluate the proposed method using the GibLayout dataset.

Figure 8 shows the precision and recall of our method with different thresholds. The threshold is mentioned above, e.g., the distance between the GT and the reconstructed wall/door vertices should be smaller than the threshold. The precision and recall of the result in the Gibson dataset are both high, indicating that the error between the line segment and the ground truth of the result is extremely small. The method proposed in this paper utilizes mesh-based resampling to obtain a point cloud with uniform distribution, which greatly improves the accuracy and thoroughness of plane detection. Figure 9 shows that the structure of our reconstructed floorplan is similar to the GT.

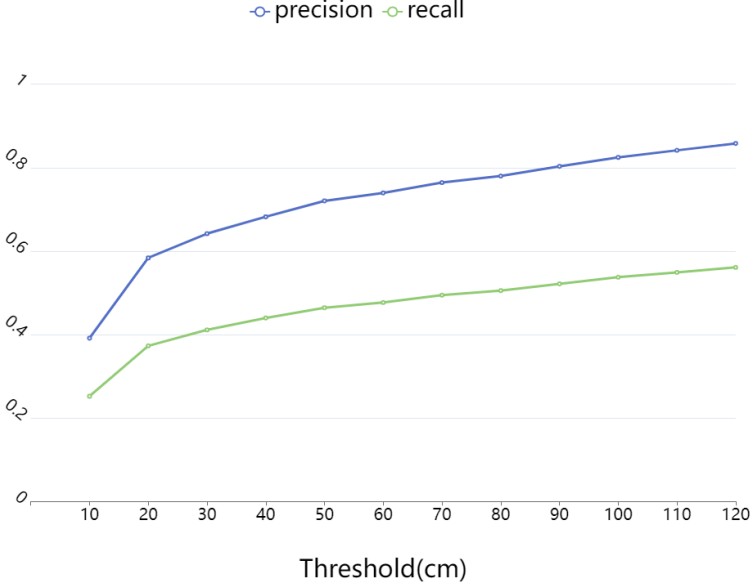

**Figure 8.** Precision and recall of our method with different distance thresholds on GibLayout Dataset.

### 4.4. Ablation Studies

In this section, we describe our ablation study to evaluate the performance of the Manhattan term in the energy function. As shown in Figure 9, the Manhattan term in the energy function penalizes the line segment with a larger tilt angle, which makes the reconstruction result close to the Manhattan world hypothesis. The energy function with the Manhattan-term constraints optimizes the selection of line segments and the connection between line segments, and improves the recall and precision of the reconstruction results. Simultaneously, adding the Manhattan term expands the topology of the floorplan, making the reconstructed topology closer to the ground truth, and the Betti error is reduced.

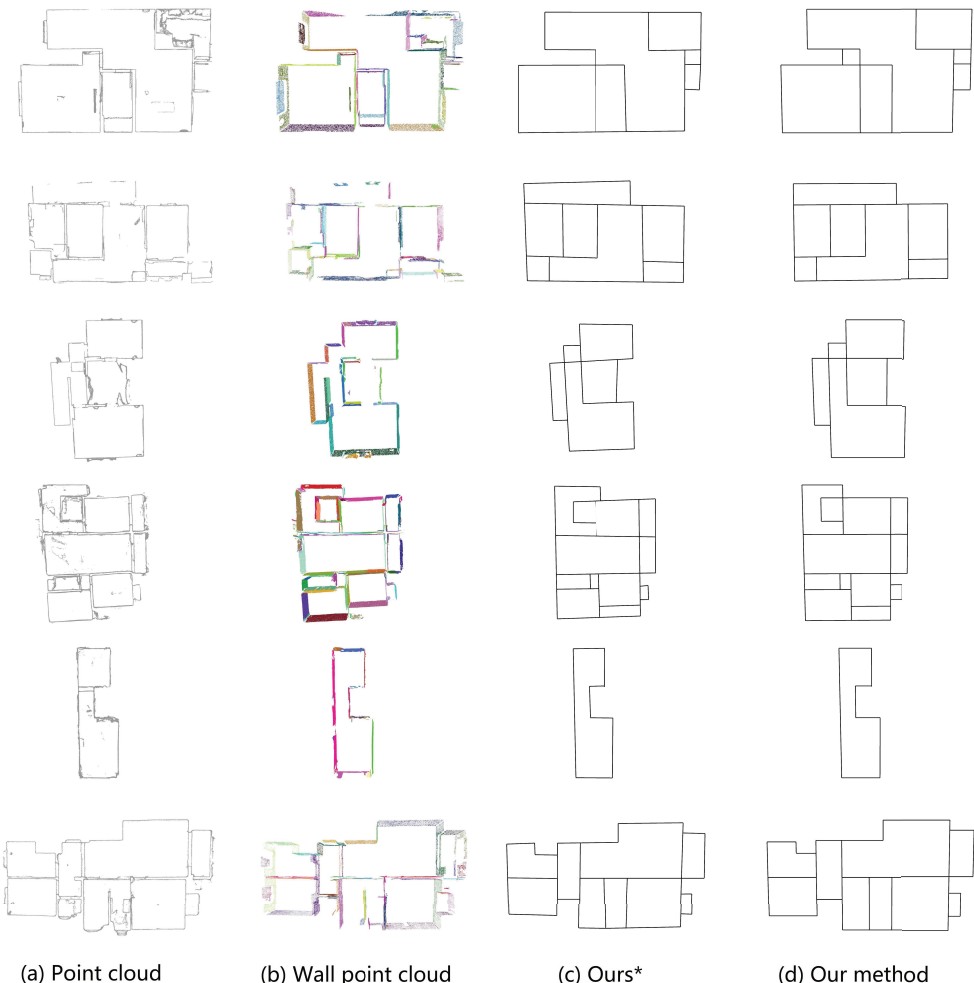

(a) Point cloud     (b) Wall point cloud     (c) Ours*     (d) Our method

**Figure 9.** Examples of floorplans predicted by our method on our datasets. From left to right, we display the density map of the point cloud, the wall point cloud, our*, and the result of our method, respectively. Our* is the non-Manhattan result of our method.

## 5. Conclusions

In this paper, we proposed an automatic indoor floorplan reconstruction framework and annotated a large high-precision dataset, named GibLayout. The proposed method utilizes the mesh-based resampling method for the normalization of input 3D models. In addition, we detected wall candidates based on normal-aware plane detection and combination. Then, the projection was performed to compress the 3D walls into 2D segments for the improvement of computational efficiency. Finally, the Manhattan term, the confidence term and the topology term were proposed to optimize the walls from the candidates via the CIP solver. Experiments show that our method generates robust and accurate floorplans, and outperforms the existing methods of FloorNet and Floor-SP.

**Author Contributions:** Conceptualisation, data curation, formal analysis, investigation, methodology, Project administration, Software, writing—original draft, Q.W.; methodology, writing—original draft, writing—review and editing, Z.Z.; methodology, formal analysis, investigation, R.C.; investigation, methodology, project administration, Software, writing—original draft, W.X.; investigation, writing—review and editing, C.Y. All authors have read and agreed to the published version of the manuscript.

**Funding:** This work was supported by the National Key Research and Development Program of China under Grant (2020YFB1406604), National Nature Science Foundation of China (61931008, 62071415), Fundamental Research Funds for the Provincial Universities of Zhejiang (GK219909299001-407), Zhejiang Province Nature Science Foundation of China (LR17F030006), Lishui Institute of Hangzhou Dianzi University(2022-001).

**Data Availability Statement:** Attached is a link to the public dataset we have generated https://github.com/W-Q-T/Giblayout (accessed on 1 March 2022).

**Conflicts of Interest:** The funders had no role in the design of the study; in the collection, analyses, or interpretation of data; in the writing of the manuscript, or in the decision to publish the results.

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
