# Peer review of "Building Floorplan Reconstruction Based on Integer Linear Programming"

_remotesensing, doi:10.3390/rs14184675_

Round 1

Reviewer 1 Report

I look forward to the final paper.

Author Response

Thanks for the comment.

Reviewer 2 Report

The authors presented a floorplan generation using deep learning and also generated a database for this aim. This part of the work is interesting, but neither the title nor the abstract can demonstrate achievements of the work. Meanwhile some important questions are posed:

- downsampling of point clouds is not always a proper way to deal with non uniform data. To which resolution you did this resampling?

- equation 5 shows this method is just defined for non Manhattan data!

- definition Di can not be correct due to the fact that it uses the main definition of Betti. What is the reference of this coefficient?

- the evaluation of the generated dataset and its accuracy were not presented

This work should be improved.

Reviewer 3 Report

In my opinion it is a very interesting and original approach. It is very well exposed and it must be rated as high scientific level.

Just to improve the presentation, the table 1 would be more efficient if converted (or additionally represented) as a chart that simplifies the interpretations of the results.

Reviewer 4 Report

The topic is interesting and can benefit the BIM industry.

However, the writing and presentation are too brief. It is hard for audiences to repeat this work based on the methods stated in section 3. The developed codes or pseudocode should be provided as well.

In the revision, the authors need to follow the Figure 1, and explain each step and process in detail. Currently, the merge & projection is missing.

Reviewer 5 Report

This research was focused on developing a method to reconstruct 2D building floorplans from 3D point cloud based on integer liner programming. A framework for floorplan reconstruction based on mesh-based point cloud normalization was also introduced. Additionally, the loose-Manhattan constraint is performed in the optimization process. The reviewer believes that the current version of the manuscript is not yet ready for publication; the authors are encouraged to consider the following comments and suggestions and revise the manuscript accordingly.

1. The authors should streamline the Introduction section and the Related Works section. The introduction section should focus on introducing the research objectives and research questions, while the Background section should focus on literature review of related work and defining the research gap. In addition, the authors should include more studies that are recent on this topic. The authors should discuss the impact of UAV or drone acquired data on 2D building floorplan reconstruction. The authors should read and cite the paper of “The Impact of Small Unmanned Airborne Platforms on Passive Optical Remote Sensing: A Conceptual Perspective”.

2. The authors should provide more discussion about the dataset used for this study. How were the point clouds collected? Were they collected using terrestrial LiDAR or airborne LiDAR? What was the point cloud’s density? What spectral bands were used? When were the point clouds collected? What were the quality level of the LiDAR point clouds? These are just example questions but the authors need to provide detailed information about the dataset that was used for this study.

3. What are RGBD images? What does D indicate? The authors did not provide adequate information about these images. How were they used for this study? This is a remote sensing paper and the authors need to provide all related information.

4. The authors need to include discussion about the ground-truth floorplans in terms of how they were collected or created. The authors need to compare not only areas but also perimeters. The authors should use appropriate metrics for this comparison. 

5. Most of the figures need to be improved. For example, in Figure 3, the reviewer has to zoom in at least 200% to be able to read. If at all possible, please create vector images for readability.

Round 2

Reviewer 2 Report

The comments and question were replied by authors. They can be acceptable.

Reviewer 4 Report

It has been improved. 

Reviewer 5 Report

The authors have addressed all my comments.